# Soft jamming of viral particles in nanopores

Léa Chazot-Franguiadakis[1], Joelle Eid[2], Gwendoline Delecourt[3], Pauline J. Kolbeck[1,4,5], Saskia Brugère[1], Bastien Molcrette[1,6], Marius Socol[2], Marylène Mougel [2], Anna Salvetti[7], Vincent Démery[1,8], Jean-Christophe Lacroix [9], Véronique Bennevault[3,10], Philippe Guégan[3], Martin Castelnovo[1] & Fabien Montel [1]✉

Viruses have remarkable physical properties and complex interactions with their environment. However, their aggregation in confined spaces remains unexplored, although this phenomenon is of paramount importance for understanding viral infectivity. Using hydrodynamical driving and optical detection, we developed a method to detect the transport of single virus in real time through synthetic nanopores. We unveiled a jamming phenomenon specifically associated with virus confinement under flow. We showed that the interactions of viral particles with themselves and with the pore surface were critical for clog formation. Based on the detailed screening of the physical and chemical determinants, we proposed a simple dynamical model that recapitulated all the experimental observations. Our results pave the way for the study of jamming phenomena in the presence of more complex interactions.

Jamming of particles is common in confined environments. This complex phenomenon has been widely studied for colloids[1–3] and nanoparticles[4,5] that are driven by flow through pores. At the scale of a single pore, the formation of a clog is controlled by pore size, flow rate, particle concentration, as well as particle–particle and particle–surface interactions (which are affected by ionic strength as described in DLVO theory)[1,2,5]. Depending on the parameters, there are different modes of clog formation: complete blocking, bridging, or standard clogging[6]. As for modeling, some studies provide empirical corrections of the advection/diffusion equations. They typically imply a modification of the adsorption kinetic constant to take into account the different types of interactions between the particles and the porous surface[4,7]. Nevertheless, the above-mentioned studies focus on a high pore/particle size ratio and the clog, therefore, corresponds to the accumulation of tens or hundreds of particles in a micrometer-scale pore. In addition, these studies only focus on synthetic particles and not on viruses, which are particular biological objects with their own complex characteristics.

Regarding viruses, aggregation can occur in solution or in contact with a surface[8]. In both cases, the aggregation is the result of interactions mainly governed by electrostatic and hydrophobic forces[9]. Indeed, majority of viruses exhibit a negative global surface charge at neutral pH[10,11] and both hydrophobic and hydrophilic residues on their surface. Consequently, the propensity of viruses for aggregation and adsorption is likely to lead to jamming in confined spaces such as nanopores[12]. This represents a technical bottleneck for electrical detection of viruses where the number of nanopore used in parallel is limited[13–15]. This limitation is partly caused by the fact that such methods require high viral concentrations, typically around $10^8$–$10^{10}$ particles/mL. This is mainly due to the use of a single pore (no parallelization of the system) and the membrane material (silicon nitride, SiN) that do not limit protein interactions[15,16]. Overall, this concentration range is not representative of the concentrations encountered in biological conditions, such as patient biofluids, where viral concentrations can vary from $10^3$ to $10^7$ particles/mL, depending on various factors such as the type and stage of infection[17].

[1]Laboratoire de Physique, UMR CNRS 5672, ENS de Lyon, Université de Lyon, Lyon, France. [2]Institut de Recherche en Infectiologie de Montpellier, UMR CNRS 9004, Université de Montpellier, Montpellier, France. [3]Institut Parisien de Chimie Moléculaire, UMR CNRS 8232, Sorbonne Université, Paris, France. [4]Department of Physics and Center for NanoScience, LMU Munich, 80799 Munich, Germany. [5]Department of Physics and Debye Institute for Nanomaterials Science, Utrecht University, 3584 CC Utrecht, The Netherlands. [6]Department of Functional Genomics and Cancer, Institute of Genetics and Molecular and Cellular Biology, UMR CNRS 7104, University of Strasbourg, Illkirch, France. [7]Centre International de Recherche en Infectiologie, UMR CNRS 5308, Université de Lyon, INSERM, Lyon, France. [8]Gulliver, UMR CNRS 7083, ESPCI Paris, Université PSL, Paris, France. [9]Université Paris Cité, ITODYS, CNRS, F-75006 Paris, France. [10]University of Evry, Evry 91000, France. ✉e-mail: fabien.montel@ens-lyon.fr

In the present work, we used hydrodynamical driving of viral particles through nanopores coupled with optical detection[18–20]. We developed a simple and sensitive approach to study virus transport in confined environments, based on synthetic membranes with high nanopore density, and which only required particle fluorescent labeling. We have previously shown that our setup can be used as a versatile tool to detect viral particles and accurately determine their concentrations with a low detection limit (<$10^5$ particles/mL) and a high precision (4% of error)[21]. Here, a different pressure and concentration regime (comparable to biofluid concentrations mentioned above[17]) was explored to study virus transport in nanopores with a small pore/particle size ratio (~1.3–16). In this particular regime, we evidenced a jamming phenomenon related to the confinement of viruses underflow. We studied the determinants of the jamming, and we proposed a physical model of this phenomenon, from which we can extract parameters related to the interactions of viruses with each other and with the pore. This phenomenological model may be applicable to a wide range of transported objects, as it does not focus on the detailed mechanism of particle interactions.

## Results

The study focused on viral particles relevant to biotechnology and pathology. We used enveloped viruses (human immunodeficiency

virus (HIV) in a non-infectious virus-like particle (VLP) form, and murine leukemia virus (MLV) in both VLP and complete virus forms), and non-enveloped particles (adeno-associated virus (AAV) and hepatitis B virus (HBV) capsids), as represented in Fig. 1A. The experimental setup involved driving fluorescently labeled viral particles through a nanoporous membrane and detecting them optically using the zero-mode waveguide effect (Fig. 1B). High pore density membranes with cylindrical nanopores were used to facilitate translocation and detection (see the "Methods" section).

### Specific jamming of viral particles under flow

Using this setup, we measured the virus translocation frequency through the nanopore as a function of pressure, for different virus concentrations. An example was depicted for HIV in Fig. 1C. The curves were normalized by a membrane-dependent prefactor $k$ (Supplementary Discussion, Supplementary Fig. 10). Two main features can be highlighted:

- No virus exited the pore until a certain pressure was reached. This critical pressure $P_c$, independent of virus concentration (Supplementary Discussion, Supplementary Table 4), was interpreted as the minimum pressure required to prevent the attachment of viruses to the pore surface, as described later in this article. For HIV particles and 200 nm diameter pores, $P_c = 50 \pm 10$ Pa (exit

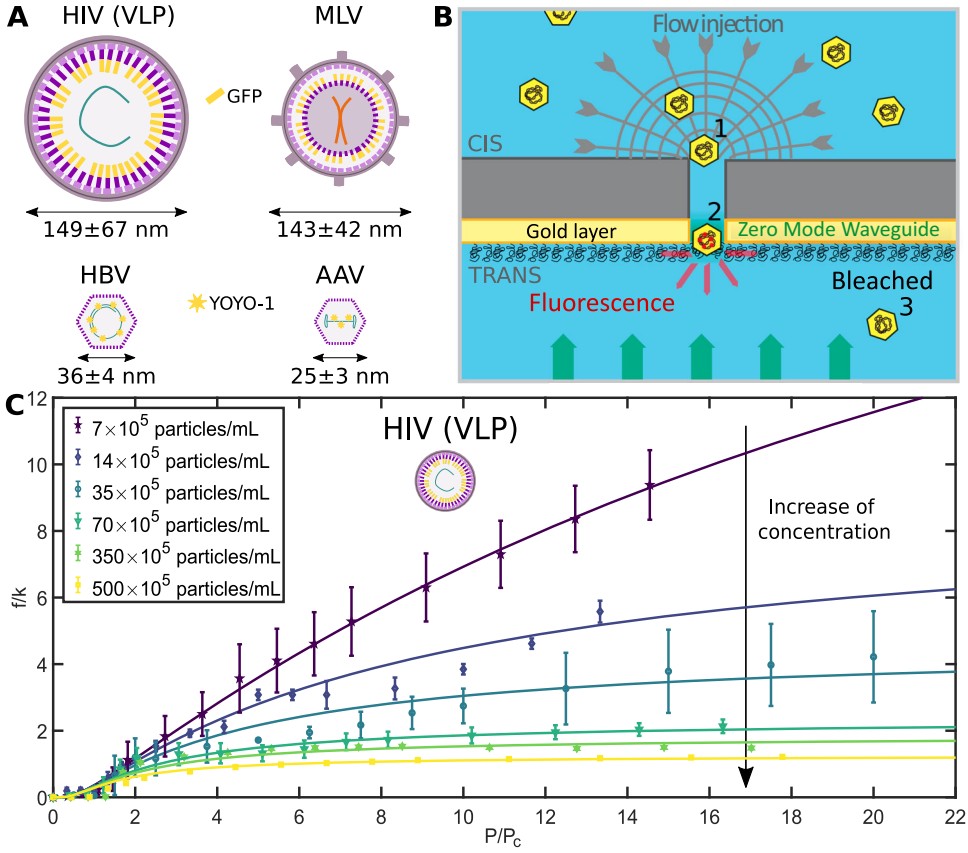

**Fig. 1 | Experimental Setup and Evidence of Virus Jamming in the Nanopore. A** Characteristics of viral particles used in this study. HIV (VLP), MLV, HBV, AAV (serotypes 8 and 9). Fluorescent labeling of particles was required: it can be achieved by genome modification (GFP labeling for HIV and MLV) or directly by adding fluorophores in the sample (YOYO-1 for AAV and HBV). Potential cellular DNA is represented in red in HIV (VLP), viral RNA in pink in MLV, and viral DNA in purple in HBV and AAV. Sizes were determined by Cryo-EM reconstruction for AAV[34]/HBV[35] and by NTA for HIV (VLP)/MLV[21]. **B** Zero mode waveguide setup for virus translocation through nanopores. The *cis* chamber was connected to a pressure controller and it contained the fluorescently labeled viral particles. Upon pressure application, the particles were transported through the nanopore in the

*trans* chamber and illuminated as soon as they crossed the evanescent field region at the end of the pore. Then, they were unfocused and bleached. **C** Evolution of the translocation frequency as a function of pressure for HIV VLP particles at different concentrations. $f$ was the translocation frequency divided by a membrane-dependent prefactor $k$. $P$ was the applied pressure divided by a critical pressure $P_c$ ($50 \pm 10$ Pa). Each curve was individually normalized. At high pressure, the increase in concentration led to a decrease in translocation frequency. Continuous color lines were fitted by the virus jamming model developed in this article (Eq. (4)). Pore diameter 200 nm. Experimental errors were the standard error of the mean, and for each experimental series, there were $N = 36$ technical replicates.

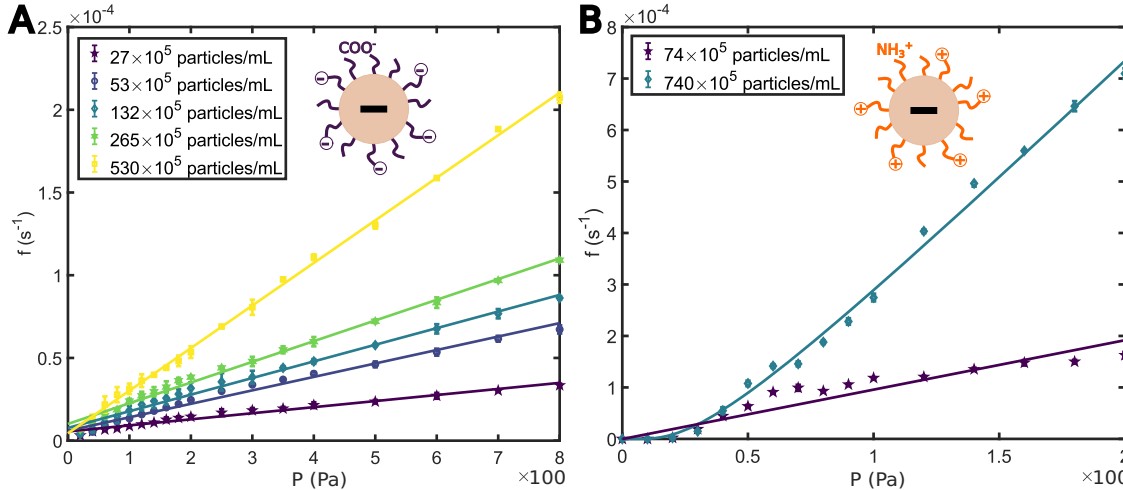

**Fig. 2 | Nanobeads Transport through Nanopores. A** The translocation frequency as a function of pressure for polystyrene nanobeads functionalized with carboxyl groups at different concentrations. Linear regimes were observed (continuous color lines) contrary to viral particles. $R_{beads} = 38 \pm 11$ nm, measured by DLS. Pore diameter 200 nm. **B** Translocation frequency as a function of pressure for polystyrene nanobeads functionalized with amino groups at different concentrations. Presence of a critical pressure (below which no nanobeads were observed) followed by a linear regime. Continuous color lines were fitted by $k^{-1}\frac{P_c}{P}\exp(\frac{P_c}{P})$ (see $\tau_2$ in Eq. (4)). $R_{beads} = 85 \pm 25$ nm, provided by the supplier. Pore diameter 400 nm. For **A** and **B** Experimental errors were the standard error of the mean, and for each experimental series, there were $N = 24$ technical replicates.

experiments). Above this critical pressure, a progressive increase in the translocation frequency was observed.

- The translocation frequency saturated at high pressures, resulting in the apparition of a pressure-independent frequency regime (frequency plateau). As the concentration increased, the frequency plateau decreased, which constituted the hallmark of a jamming phenomenon.

Remarkably, even for high pressures and high concentrations, a non-zero translocation frequency was observed, and there was no complete pore clogging (Supplementary Discussion). For sufficiently high concentrations, frequency versus pressure curves became indistinguishable from each other. Therefore, these experiments revealed a soft jamming phenomenon related to the confinement of viruses underflow, which was observed for all viruses tested (Fig. 1A). Moreover, in the experimental conditions used, we evidenced that this phenomenon was specific to viruses, comparing their transport to that of simple nanobeads. Indeed, we performed experiments with fluorescent polystyrene nanobeads (Supplementary Methods) in a concentration range similar to the one used for viruses. These nanobeads exhibited on their surface two types of chemical groups, as shown in Fig. 2. First, we relied on nanobeads functionalized with carboxyl groups ($R_{beads} = 38 \pm 11$ nm) with limited attractive interaction with the pore surface. For these nanobeads, we observed a linear evolution of the translocation frequency with pressure and concentration (Fig. 2A). The linearity supported that the transport through a nanopore was dominated by advection (Peclet number, $P_e > 1$; Supplementary Discussion) and that the flow in our system verified the Poiseuille law. Complementary experiments with other carboxyl nanobeads and pore sizes were performed and showed similar linear behavior (Supplementary Discussion, Supplementary Fig. 4). Secondly, we carried out similar experiments but with nanobeads functionalized with amino groups ($R_{beads} = 85 \pm 25$ nm), which induced interaction with the pore surface. For these nanobeads, we observed the appearance of a critical pressure similar to the one observed for viruses (Fig. 2B). However, there was no saturation at high pressures, and the increase in concentration was accompanied by an increase in translocation frequency. In both cases (carboxyl and amino nanobeads), there was no attractive interaction between the nanobeads themselves, but only interaction of

the nanobeads with the pore surface in the case of the amino functionalization. These elements suggested that the jamming phenomenon depended on interaction with the pore surface, but that this interaction alone was not sufficient to account for the phenomenon.

## Characterization of the viral clog

We then examined the jamming phenomenon observed for viruses in more detail by conducting different experiment types. First, we performed similar experiments to the ones described in Fig. 1B and C, but we looked at virus entry in the nanopore instead of the exit. For this purpose, viruses were introduced into the trans chamber while the pressure controller was working in suction (Fig. 3A; Supplementary Discussion). From a visual point of view, the videos acquired showed that there was no aggregation of viruses at the entrance or exit of the nanopores. Consequently, we assumed that the viral clog was located inside the nanopore (Supplementary Discussion, Supplementary Fig. 6). Moreover, in entry experiments (Fig. 3A), we highlighted the absence of a critical pressure (for HIV particles and 200 nm diameter pores, $P_c = 10^{-3}$ Pa, in entry experiments). Right from the beginning, the translocation frequency evolved linearly up to a certain point, afterwards it saturated in a similar way as in exit experiments. We interpreted the absence of critical pressure by the fact that there was no adhesion of viruses to the pore at the channel entrance.

In addition, to explore the saturation phenomenon observed in Fig. 1C, we probed the transient state of clog formation. For that, we used a high pressure ($10^4$ Pa) to remove the viral clog that had formed in the central channel, as represented in Fig. 3B. At $t = 0$, the central channel was empty, and the clog formation was followed during 5 min at a fixed pressure of 800 Pa (frequency saturation regime). The translocation frequency decreased exponentially which we related to clog formation. A characteristic duration can, therefore, be extracted, which did not depend on the steady-state pressure (Supplementary Discussion, Supplementary Fig. 7) and was about $t_{clog} = 38$ s for HIV particles. If we considered that, during clog formation, viruses were transported on a distance equivalent to pore length ($L = 10$ μm), it appeared that advection alone cannot account for the measured duration. Indeed, advection duration ($t_{adv} = \frac{L}{v} = \frac{8\eta L^2}{\Delta P R^2} \sim 0.1$ s, with: $R$, pore radius; $\eta$, water viscosity; and $\Delta P$ applied pressure) was much shorter than the observed duration. Actually, the measured duration found was of the order of magnitude of a diffusive phenomenon:

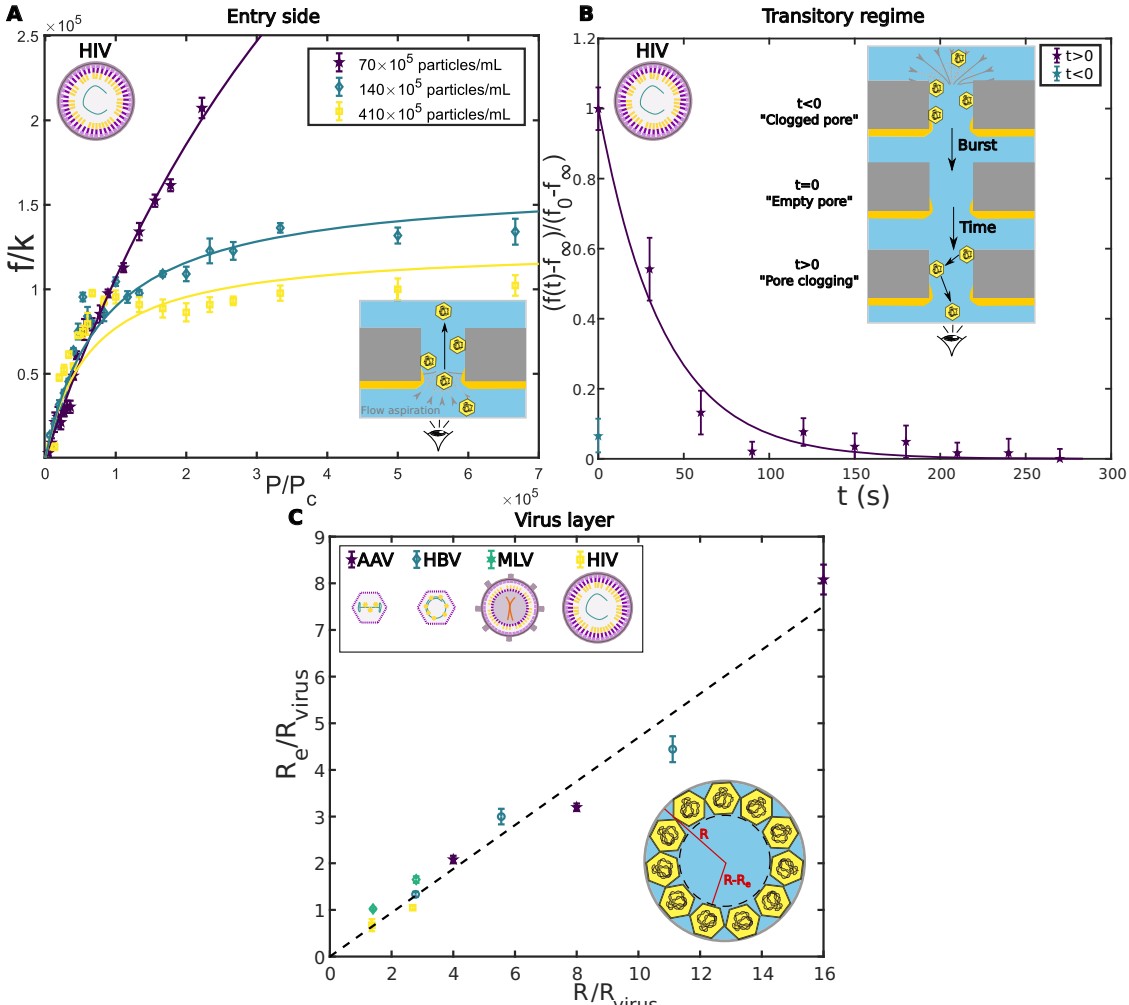

**Fig. 3 | Virus Clog Characterization. A** Entry of viruses in the nanopore. Evolution of the translocation frequency as a function of pressure for HIV particles at different concentrations (entry side). $f$ was divided by a membrane-dependent prefactor $k$. $P$ was divided by a critical pressure $P_c$ ($10^{-3}$ Pa). Continuous color lines were fitted by the virus jamming model developed in this article. Pore diameter 200 nm. **B** Dynamic of the viral clog. Evolution of the frequency contrast ($\frac{f(t)-f_\infty}{f_0-f_\infty}$) as a function of time after clog removal, for HIV particles ($95 \times 10^5$ particles/mL). $f_0$ and $f_\infty$ were, respectively, the translocation frequencies at $t = 0$ and $t = 300$ s. By using a high pressure ($10^4$ Pa during 2 min), the virus clog inside the pore was removed at

$t = s$ and the dynamic of formation of the clog was followed for 300 s. The continuous color line was a fit by a decreasing exponential ($\exp(-\frac{t}{t_{clog}})$, with $t_{clog} = 38$ s). Pore diameter 200 nm. **C** Virus layer. Evolution of the thickness of virus layer normalized by virus radius as a function of pore radius normalized by virus radius. Pore diameter 100, 200, and 400 nm. The black dotted line corresponds to a linear fit with a proportionality coefficient of 0.5. For **A**–**C**: Experimental errors were the standard error of the mean, and for each experimental series there were $N = 24$ technical replicates.

$t_{diff} = \frac{L^2}{D_{virus}} = 38 \pm 5$ s. $D_{virus}$ is the diffusion coefficient of the virus: $D_{virus} = \frac{k_B T}{6\pi\eta R_{virus}}$, with $k_B$, the Boltzmann constant, and $T$ the temperature. A more detailed discussion on the competition between advection, diffusion and binding can be found in Supplementary Discussion).

Once the steady state was reached and the clog formed, we sought to characterize the structure of the clog, notably the number of viruses jammed in the pore. For that, a clog was established by flowing viruses into the nanopore in the frequency saturation regime (800 Pa, for 200 nm pores) for 20 min (steady-state reached). Then, we used the transport properties of double-stranded (ds) DNA to probe the structure of the clog. Indeed, dsDNA translocation through nanopores is well described by the suction model[18–20]. The latter predicts that the translocation of DNA through nanopores is similar to the crossing of an energy barrier. This barrier comes from the competition between the work done by the flow onto the DNA and an entropic term that comes from the confinement of the DNA inside the nanopore. The translocation frequency for DNA, $f_{DNA}$, could therefore be expressed as: $f_{DNA} = f_{suc} \frac{P}{P_{suc}} \exp(-\frac{P_{suc}}{P})$, with $P_{suc}$, the critical pressure at which the

energetic cost of the translocation is balanced by the energy injection from the pressure gradient and $f_{suc}$, a prefactor corresponding to the translocation frequency for $P = P_{suc}$. Moreover, if we assumed that the transport through the nanopore clogged with viruses, verified the Poiseille law (Supplementary Discussion), $P_{suc}$ can be directly related to the radius of the nanopore: $P_{suc} = J_c R_h$, where $J_c = \frac{k_B T}{\eta}$, corresponds to the critical flow that is sufficient to induce polymer translocation in a nanopore[18–20] and $R_h = \frac{8\eta L}{\pi R^4}$ is the pore hydraulic resistance. Therefore, by comparing the value of $P_{suc}$ in the case of a naked nanopore or a nanopore clogged with viruses (Supplementary Discussion, Supplementary Fig. 8), we can extract the virus thickness inside the nanopore, $R_e$. Assuming that viruses formed a compact layer on the border of the pore we obtained:

$$R_e = R - \frac{R}{\left(\frac{P_{suc}^{virus}}{P_{suc}}\right)^{1/4}} \tag{1}$$

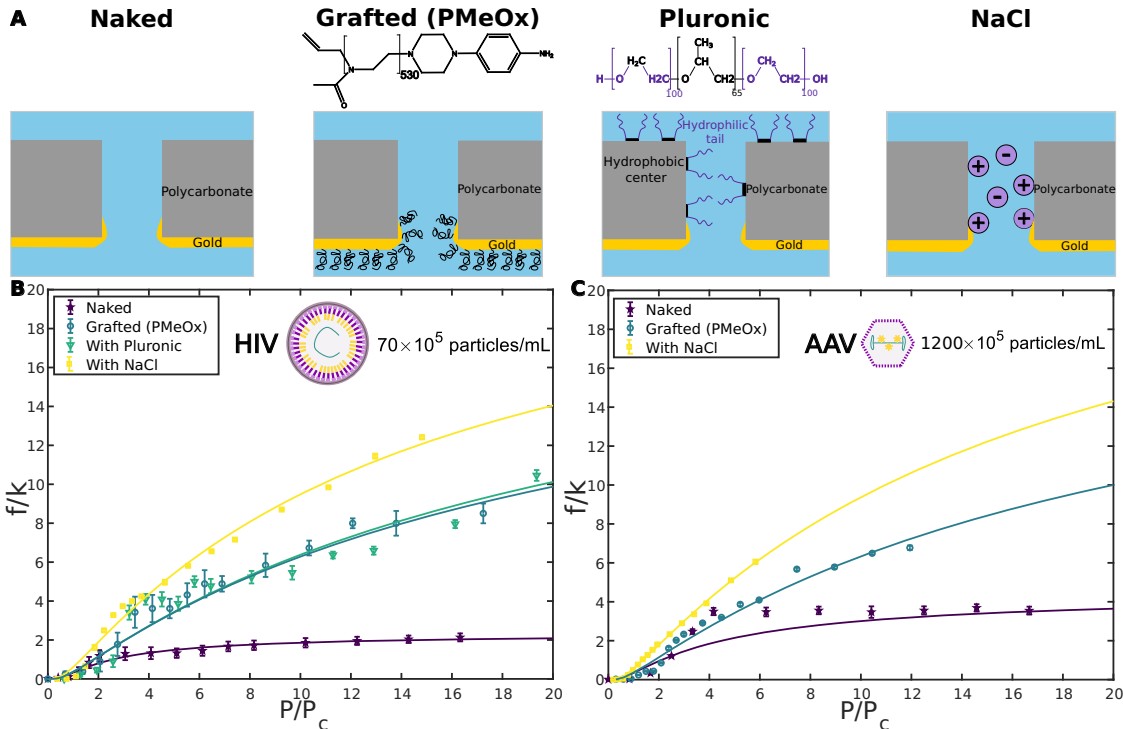

**Fig. 4 | Chemical Environment Modifications. A** Schematic representation of the nanopore environment. From left to right: naked nanopore; nanopore grafted with synthetic polymers (PMeO$_x$, polymerization degree of 530) on the gold side; nanopore passivated with Pluronic; nanopore in the presence of 150 mM NaCl. **B** HIV translocation for different nanopore environments. Evolution of normalized translocation frequency as a function of normalized pressure for HIV particles for different nanopore environments: naked, grafted (PMeO$_x$), passivated (Pluronic), with NaCl (150 mM). The translocation frequency was higher for grafted and passivated membranes and even higher with salt compared to naked membranes. The saturation was also less marked in these cases. **C** AAV translocation for different nanopore environments. Evolution of normalized translocation frequency as a function of normalized pressure for AAV-9 particles for different nanopore environments: naked, grafted (PMeO$_x$), with NaCl (150 mM). Normalized frequency and pressure were obtained for HIV. For **B** and **C**, the Pore diameter 200 nm. Experimental errors were the standard error of the mean, and for each experimental series, there were $N = 24$ technical replicates. Continuous color lines were fit by the virus jamming model developed in this article.

---

With $P_{suc}$ and $P_{suc}^{virus}$, critical pressures from the suction model, respectively, for a naked nanopore and a nanopore clogged with viruses. Similar experiments have already been performed with our system, to measure the thickness of poly(N-isopropylacrylamide)[22] or of poly(2-alkyl-2-oxazoline)s[20] grafted onto gold nanoporous membrane. The evolution of the ratio $\frac{R_e}{R_{virus}}$ as a function $\frac{R}{R_{virus}}$ was plotted for different viruses and pore diameters in Fig. 3C. The data were fitted by a linear relation that provided $R_e = 0.5R$ for all viruses and pore diameters. We concluded from this measurement that viruses occupied a large part of the pore. A more detailed investigation would be necessary to decipher the precise structure of the clog.

### Influence of the chemical environment

After describing the jamming, the effect of the chemical environment on this phenomenon was investigated. Assuming that the jamming was partly related to the attractive interaction between viruses and the pore, we decided to modify the pore surface (Fig. 4A). We either grafted synthetic polymers (poly(2-methyl-2-oxazoline)s, PMeOx) onto nanoporous membranes[20,23], or passivated the surface using Pluronic, which is a non-ionic surfactant (Supplementary Methods). Compared to virus translocation through a naked membrane, transport through a membrane grafted with PMeOx or passivated with Pluronic was increased. Fig. 4B and C showed, respectively for HIV and AAV, a higher translocation frequency through a grafted or passivated membrane compared to a naked membrane. The saturation was also less pronounced in the case of the modified membranes. However, the three curves exhibited the same profile with the existence of a critical pressure, followed by an increase and then a

saturation of the translocation frequency. It can also be noticed that grafting or passivation had the same influence on the translocation frequency. These elements indicated a decrease in the interaction of viruses with the pore surface and, thus, a transport that was eased. This can be explained partly by steric hindrance effects due to the presence of polymers, which prevent viruses from reaching the pore surface.

Moreover, we also tuned the ionic strength by adding salt to virus solutions (Fig. 4A). The addition of 150 mM NaCl to HIV and AAV solutions led to an increase in the translocation frequency (Fig. 4B and C). The behavior was similar to that observed for the modified membranes, except that the translocation frequency was even higher and the saturation was less pronounced. Complementary experiments with higher NaCl concentrations (300 and 600 mM) were performed, and resulting curves exhibited the same behavior (Supplementary Discussion, Supplementary Fig. 9). In this case, the transport of viruses was facilitated by electric charge screening that prevented electrostatic interactions of viruses with each other or with the pore. Indeed, viruses also present charged groups on their surface and usually exhibit a negative zeta potential at neutral pH (−4.0 mV for HIV[10] and −10 mV for AAV-9[11]). Moreover, even if there is no consensus regarding the salt effect on proteins' attachment to surfaces, a decrease of attachment has already been reported for BSA adhesion to polymer membranes while increasing salt concentration[24].

Consequently, the jamming phenomenon was affected by the modification of the chemical environment and depended on electrostatic and hydrophobic interactions.

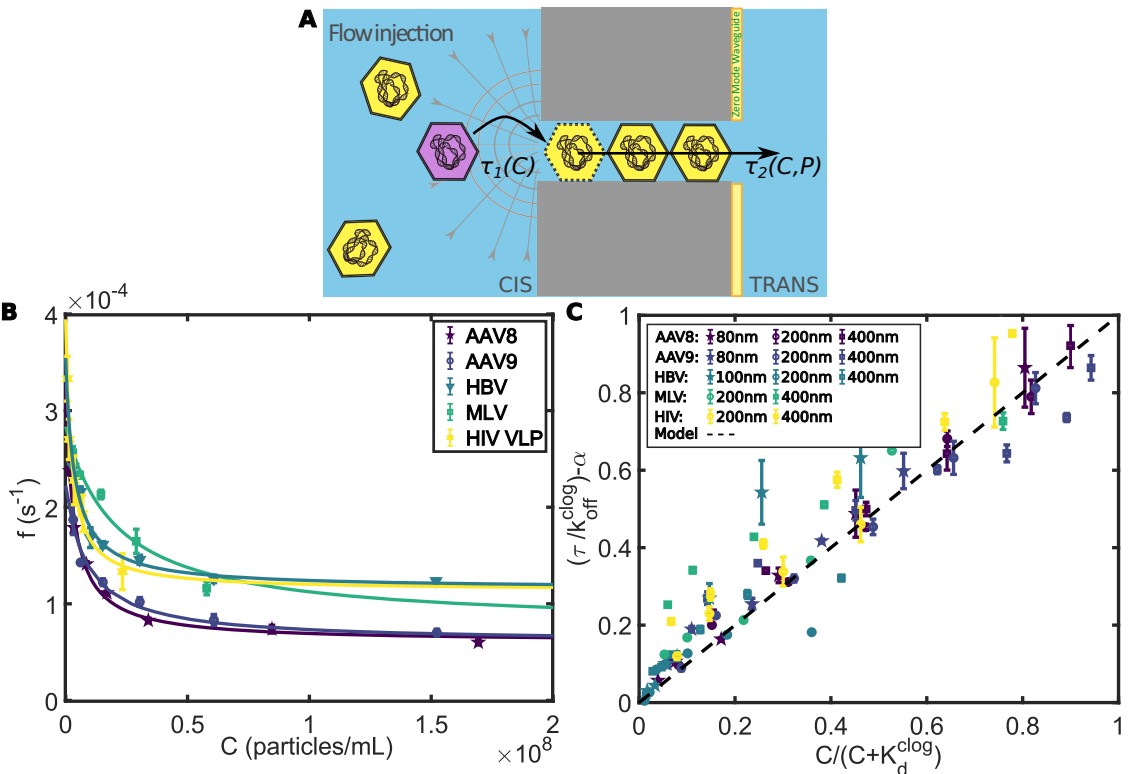

**Fig. 5 | Soft Jamming Modeling. A** Modeling of virus jamming phenomenon. Total translocation duration of viruses depended on the duration of entry in the nanopore ($\tau_1(C)$) and on the transport duration through the nanopore ($\tau_2(C, P)$). **B** Evolution of the translocation frequency as a function of virus concentration for different viral types. Continuous color lines were fit by the virus jamming model developed in this article. Pressure fixed at 800 Pa. Pore diameter 200 nm. **C** Master curve of the model. Evolution of $\frac{\tau}{k_{off}^{clog}} - \alpha$ (with $\alpha = \frac{P_c}{P} \exp(\frac{P_c}{P})$) as a function of $\frac{C}{C + K_d^{clog}}$ for different viruses (AAV-8/AAV-9/HBV/HIV/MLV) and different pore diameters (80/100/200/400 nm). The dashed line corresponds to the jamming model proposed in this article. For **B** and **C**: Experimental errors were the standard error of the mean, and for each experimental series, there were $N = 18$ technical replicates.

## Soft jamming modeling

Finally, the detailed screening of the physical and chemical determinants allowed us to converge toward a model for the flow-driven jamming of viruses in nanopores (Supplementary Discussion). The physical model, that we proposed, depicted the phenomenon as an aggregation under flow. We assumed that advection to the pore was not a limiting step. This assumption has been used in another model, such as the suction model for the transport of DNA molecules[18]. Here, the barrier is limited not by a confinement energy barrier but by the accessibility of the site as in Langmuir isotherm models. Hence, the translocation time was dominated by this barrier while the time to reach the pore was assumed to be negligible. The duration of a virus passage through a nanopore can, therefore, be decomposed into two independent timescales: $\tau_1$ (virus entry duration) and $\tau_2$ (virus translocation duration through the pore), as schematized in Fig. 5A.

We assumed that the entry duration ($\tau_1$) was related to the interaction of viruses with the clog. We made the hypothesis that entry duration depended only on the probability of the pore being occupied by other viruses ($P_{\text{virus}}$). Supposing that the equilibrium between viruses in solution and those associated with the clog was given by Langmuir equilibrium, the entry time can be expressed as

$$\tau_1(C) = (k_{\text{off}}^{\text{clog}})^{-1} P_{\text{virus}} = (k_{\text{off}}^{\text{clog}})^{-1} \frac{C}{C + K_d^{\text{clog}}}, \quad (2)$$

with $C$, the particle concentration in the upstream chamber, $k_{\text{off}}^{\text{clog}}$ the characteristic off-rate at which the virus detached from the clog, and $K_d^{\text{clog}}$ the dissociation constant between the virus and the clog.

Concerning the virus translocation duration through the pore ($\tau_2$), we took into account the potential adhesion of viruses to the nanopore

channel. We hypothesized that in the stationary regime, inside the nanopore, the concentration of viruses, $C_n(x, t)$, can be described as an interplay between adhesion to the pore surface and advection by the driving flow. Concentration of viruses in the nanopore may then be written as

$$C_n(x, t) = C \exp\left(-\frac{k_{\text{on}}^{\text{pore}}}{v} x\right), \quad (3)$$

with $C$, the particle concentration in the upstream chamber (at $x = 0$), $v$, the flow speed and $k_{\text{on}}^{\text{pore}}$, the characteristic on-rate at which a virus sticks to the nanopore central channel.

To introduce our experimental control parameter, the pressure difference between the two chambers ($P$), we assumed that the transport through a nanopore was predicted by the Poiseuille law (Supplementary Discussion). In our cylindrical geometry, $v = \frac{P}{\pi R^2 R_h}$, with $R_h = \frac{8\eta L}{\pi R^4}$, where $R$ and $L$ are, respectively, the radius and the length of the pore, and $\eta$ is the water viscosity.

Finally, the translocation frequency of a virus through a nanopore can be written as

$$f(C, P) = \frac{1}{\underbrace{\tau_1(C)}_{\text{entry}} + \underbrace{\tau_2(C, P)}_{\text{exit}}} = \frac{k}{\frac{k}{k_{off}^{clog}} \frac{C}{C + K_d^{clog}} + \frac{P_c}{P} e^{\frac{P_c}{P}}}, \quad (4)$$

introducing a prefactor $k = \pi k_{\text{on}}^{\text{pore}} \text{CL} R^2$ and a critical pressure $P_c = \frac{8\eta k_{\text{on}}^{\text{pore}} L^2}{R^2}$.

At low pressures, before the saturation phenomenon, the experimental measurements of the translocation frequency as a

function of pressure were readily fitted by $f = \frac{1}{\tau_2}$ (Supplementary Discussion, Supplementary Fig. 10). It allowed us to extract the critical pressure and the related characteristic on-rate, $k_{\mathrm{on}}^{\mathrm{pore}}$. For 200 nm pores, the mean value for viruses varied from 20–60 Pa for $P_c$ and 0.3–0.8 s$^{-1}$ for $k_{\mathrm{on}}^{\mathrm{pore}}$. Furthermore, $\tau_2$ can account for the presence of a critical pressure and the absence of saturation, observed with amino nanobeads (only interacting with the pore surface).

To access both parameters $k_{\mathrm{off}}^{\mathrm{clog}}$ and $K_{\mathrm{d}}^{\mathrm{clog}}$, we carried out experiments in which the translocation frequency of viruses as a function of concentration was directly measured. For that, we set a fixed pressure (800 Pa for 200 nm pores) and varied the concentration of viruses introduced in the system, $C$. We extract $k_{\mathrm{off}}^{\mathrm{clog}}$ and $K_{\mathrm{d}}^{\mathrm{clog}}$ from the corresponding curves (Fig. 5B). For 200 nm pores, the mean value for viruses varied from $0.8$–$1.7 \times 10^{-4}$ s$^{-1}$ for $k_{\mathrm{off}}^{\mathrm{clog}}$ and $1$–$30 \times 10^{-14}$ M for $K_{\mathrm{d}}^{\mathrm{clog}}$.

Overall, the experimental data for different viruses and conditions (membrane naked, grafted, or passivated; higher ionic strength) were successfully fitted by our jamming model (continuous lines in Figs. 1C, 3A, 4B and C). Extracted parameters were related to the interaction of viruses with the pore surface ($k_{\mathrm{on}}^{\mathrm{pore}}$) and with the clog ($k_{\mathrm{off}}^{\mathrm{clog}}$ and $K_{\mathrm{d}}^{\mathrm{clog}}$). The independence of the two durations ($\tau_1/\tau_2$) was also analyzed in Supplementary Discussion (Supplementary Fig. 11) and confirmed the validity of the model. Finally, we built a master curve that related $\frac{\tau}{k_{\mathrm{d}}^{\mathrm{clog}}} - \alpha$ (with $\alpha = \frac{P_c}{P} \exp(\frac{P_c}{P})$) as a function of $\frac{C}{C + K_{\mathrm{d}}^{\mathrm{clog}}}$ (Fig. 5C). It gathered all the data from the different viral types and pore diameters. The data collapse on the master curve (dashed line) emphasized the strength of our theoretical approach.

## Discussion

Altogether, these results shed light on a soft jamming phenomenon due to the confinement of viruses underflow in the nanopore. Significantly, we highlighted the general nature of this phenomenon for all types of viruses that were studied, but not for nanobeads. We interpreted the absence of jamming for nanobeads by the lack of attractive interactions between particles, contrary to viruses. The impact of the chemical environment on the jamming was also studied by modifying the nanoporous surface and increasing the ionic strength. It revealed that both hydrophobic and electrostatic interactions held major roles in clog formation. Finally, we proposed a simple dynamic model of the jamming that recapitulated experimental data for the different conditions.

Regarding the values of the parameters extracted from the model, the effective concentration in the pore ($C_n$) has to be taken into account to interpret the dissociation constant ($K_{\mathrm{d}}^{\mathrm{clog}}$). Considering that $R_e = 0.5R$, the volume occupied by viruses in the nanopore came as: $V_{\mathrm{virus}}^{\mathrm{pore}} = \pi R^2 L (1 - (1 - \frac{R_e}{R})^2)$. We can, therefore, roughly estimate that: $C_n = \frac{V_{\mathrm{virus}}^{\mathrm{pore}}}{4/3\pi R_{\mathrm{virus}}^3 \times \pi R^2 L} \sim 10^{14} - 10^{17}$ particles/mL. A factor of $10^8$ can be deduced between the saturation concentration upstream of the pore ($C \sim 10^8$ particles/mL) and the internal concentration. With this corrective factor, we obtained $K_{\mathrm{d}}^{\mathrm{clog}} \sim 1.5 \times 10^{-5}$ M. This value is close to literature values for protein–protein interactions ($K_{\mathrm{d}} \sim 10^{-5}$–$10^{-4}$ M between different amino acids[25]). The dissociation constant should, therefore, be lower for proteins with several amino acids, as in viruses. Lastly, the characteristic on-rate at which a virus stick to the nanopore surface ($k_{\mathrm{on}}^{\mathrm{pore}} \sim 0.3 - 0.8$ s$^{-1}$) can be interpreted as the crossing of an energy barrier before the adhesion of the virus to the pore surface: $k_{\mathrm{on}}^{\mathrm{pore}} \propto e^{\frac{-\Delta F^*}{k_B T}}$. The energy barrier $\Delta F^*$ can, therefore, be estimated at $6$–$12 k_B T$. The value of $k_{\mathrm{on}}^{\mathrm{pore}}$ was also compared to Bio-Layer Interferometry measurements. We relied on a hydrophobic substrate (aminopropyl silane) and measured its interaction with AAV-8 (Supplementary Discussion, Supplementary Fig. 12). For the $k_{\mathrm{on}}$ with the

surface, we obtained a value of $0.1 \pm 0.05$ s$^{-1}$ in close agreement to the one obtained by nanopore.

Overall, by proposing a physical and quantitative description of virus jamming, our approach demonstrates the importance of self-interaction between particles in the jamming transition. It offers a new approach to characterizing surface states, providing valuable insight for studying the influence of drugs on viral particle and their interactions. Similarly, it can also be used to study other types of biological particles with a similar size and biological composition to viruses, but that exhibit different surface receptors, such as extracellular vesicles. It also opens new possibilities to engineer controlled aggregation of patchy nanoparticles under flow confinement, promising advancements in materials science and biotechnology.

## Methods
### Virus preparation

The viral particles used in this study are of interest either for biotechnology applications or simply as pathology generators, as detailed in Fig. 1A. More precisely, we relied on two enveloped viruses: the human immunodeficiency virus (HIV) in a non-infectious virus-like particle (VLP) form; and the murine leukemia virus (MLV) in the form of both VLP and complete enveloped virus. VLP structurally mimics the native virus but is devoid of the viral genome and is thus a promising candidate for vaccination[26]. For these retroviruses, fluorescent labeling relied on green fluorescent protein (GFP, $\lambda_{\mathrm{exc}} = 488$ nm/$\lambda_{\mathrm{em}} = 507$ nm) insertion into the viral genome and expressed as Gag-GFP fusion[27], which did not alter the properties of the virus[28,29]. Besides, we used two non-enveloped particles: either derived from recombinant Adeno-Associated Virus (AAV, serotypes 8 or 9), which have gained interest as a recombinant vector for genetic therapies and vaccination over the past years[30,31]; or Hepatitis B Virus (HBV) capsids derived from detergent-treated HBV particles. These capsids were both fluorescently labeled using intercalating fluorophore YOYO-1 (Molecular Probes, $\lambda_{\mathrm{exc}} = 491$ nm/$\lambda_{\mathrm{em}} = 509$ nm) which targets viral DNA. All particles were depicted in Fig. 1A and were used in Tris-EDTA buffer (10 mM of Tris–KCl, 1 mM of EDTA, pH = 7.5). Further details on viral particle production, characterization, and fluorescent labeling are available in Supplementary Methods.

### Experimental setup

The experimental setup has been developed based on a previous system in which viral particles are driven through a nanoporous membrane and detected optically using the Zero-Mode Waveguide effect[20,21]. More specifically, the viral particles are fluorescently labeled and then injected into the cis chamber of our setup. As shown in Fig. 1B, hydrodynamical driving, induced by a pressure difference (microcontroller MFCS, Fluigent) between the two sides of the nanoporous membrane, is used to induce translocation of particles from cis to trans side.

Contrary to electrical detection methods, which rely on the use of a single pore, the membranes employed are commercially available track-etched membranes (Whatman, GE-Polycarbonate), produced through heavy ion irradiation, with high pore densities ($1$–$6 \times 10^8$ pores/cm$^2$, which corresponds to $10^4$–$10^5$ pores in parallel in the field of view). The nanoporous membranes exhibit cylindrical nanopores of controlled diameters (80–400 nm) with thicknesses of 6–10 μm and are coated with a 50 nm-thick gold layer on the trans side (Supplementary Methods). They are designed to exhibit a low retention of proteins contrary to SiN membranes. Furthermore, we also passivated the surface with Fetal Bovine Serum to further limit the potential interaction between viruses and pores. Consequently, in our experiments the concentration typically varied between $10^5$–$10^7$ particles/mL, thus limiting irreversible blockages (Supplementary Discussion).

After crossing the membrane, successful translocation events were detected by a Zero-Mode Waveguide (ZMW) setup. Briefly, the nanoporous array was illuminated by a laser beam ($\lambda_{exc}$ = 473 nm) on the trans side. The gold layer inhibited the propagation of light through the membrane and also induced an enhancement of the electromagnetic field in the nanopore at a depth equivalent to the radius of the pore[32,33]. This effect enabled to optically separate the fluorescence signal of translocating particles from particles in the bulk[18–21] (Supplementary Methods). The viral particles used in this study were successfully detected at the exit of nanopores with a signal-to-noise ratio superior to 2[21].

## Reporting summary

Further information on research design is available in the Nature Portfolio Reporting Summary linked to this article.

## Data availability

Microscopy movies generated in this study have been deposited in the Zenodo database under accession code: https://zenodo.org/records/11185140?token=eyJhbGciOiJIUzUxMiJ9.eyJpZCI6ImFlMmY3NGVhLWQ2MWEtNGI2OS04N2IyLTU2MTZkMWYxYTJkZSIsImRhdGEiOnt9LCJyYW5kb20iOiIyMjJiM2E2E2YjkxOTIkNjNjMjNjMGFlOWQ3NzU4NGM5MSJ9.Ya-7Ord0EQH3M0B-LJBtDCthvrgASjtn4rtvTAKUz6ysyWDPZJsKdYw16eVtI1RQQbhwFQtd27cVnI3upVDFYA. Source data are provided with this paper.

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

## Acknowledgements

The authors thank Cendrine Faivre-Moskalenko, Pierre Illien, and Julian Talbot for their fruitful comments and discussions. We thank the platform SFR Biosciences, where the BLI experiments were performed. We thank the Laboratoire de Chimie of Ecole Normale Supérieure de Lyon and Fréderic Lerouge, who helped in conducting the Zeta

Potential and DLS measurements. This work was supported by the Centre National de la Recherche Scientifique under the 80 Prime project "NanoViro" (F.M.).

## Author contributions

F.M., P.G. conceived the project. G.D., V.B., and L.C.-F. performed polymer synthesis, characterization, and electro-grafting on nanoporous membranes. A.S. produced the AAV and HBV capsids. J.E., M.S., and M.M. produced the HIV and MLV viral particles. L.C.-F., P.J.K., S.B., B.M., and G.D. performed nanopore experiments. L.C.-F. and F.M. performed data analysis. L.C.-F., F.M., V.D., and M.C. performed theoretical modeling. F.M., M.C., P.G., V.B., and J.-C.L. supervised the work. L.C.-F., M.C., and F.M. wrote the manuscript. All authors contributed to the ideas and reviewed the manuscript.

## Competing interests

The authors declare no competing interests.
