## [Peer Review File · Nature Communications]

REVIEWER COMMENTS

Reviewer #2 (Remarks to the Author):

The model equations constructed in this paper are in good agreement with experimental results, and the paper is considered to be of high engineering and industrial value in terms of adsorption process control in nanopore-based bionanoparticle measurements, not only for viruses but also for bacteria, extracellular vesicles, and others.

The model equations are in good agreement with the phenomenon, but do not include factors related to the mechanism. In other words, the model is valid whether the adsorption mechanism of the virus on the nanopores is electrostatic, hydrophobic interaction, or other causes. This is what makes the model easy to use and valuable, but it is also what limits it to phenomenological aspects.

Some of the experimental results contain findings on the adsorption mechanism of viruses on nanopores, but there seems to be a slight inconsistency in the discussion in the main body and in the Appendix, which needs to be sorted out.

In summary, this paper has value in providing guidelines for analyzing and controlling the adsorption process in nanopore measurement of bionanoparticles and related measurement techniques in a broad range of applications.

Please correct the comments below on points of concern.

Although authors state that no irreversible blockage occurred in this study, the virus measurements in general electrical nanopore measurements use particle concentrations of about 10^8 - 10^{10} , and irreversible blockage commonly occurs. Any thoughts on the difference between them? Or if you can show results at such high concentrations, it would be a further contribution to nanopore research.

Line 140:

1 - 6.108pores, where '8' is probably "x". Other similar expressions are scattered throughout.

Please follow the author's instructions.

Figure 2 is not cited in the main body, please cite and explain.

There are commercially available beads of the same size and material, differing only in surface modification.

Shouldn't the verification using beads be measured with the same pore and particle size?

It would be better to use only surface modification as a parameter to clearly show the effect of electrostatic interactions.

In the evaluation of the effect of electrostatic forces using beads, it would be desirable to indicate the zeta potential of the beads and nanopore membranes, since it can clearly indicate whether the electrostatic interaction between the beads and the nanopore membrane is an attractive or repulsive force. In general, the zeta potential of amino group modified beads at neutral pH is lower than that of carboxyl group modified beads, so the magnitude of the zeta potential may be more important than the positive or negative difference. In such cases, hydrophobic interactions may be dominant.

Lines 232-238:

It is stated that when the virus was put in from the gold-coated side, the virus did not aggregate near the inlet and no critical pressure was generated. From this result, it is inferred that the cause of the critical pressure in the experiments in Section II is that the virus aggregated near the inlet when it was put in from the polycarbonate side.

On the other hand, the description in Fig. S3.1 states that no difference is observed whether the inlet side is gold or polycarbonate. Furthermore, a closer look at Fig. S3.1 shows that the critical pressure seems to occur around 0.2-0.3 mbar. Doesn't this contradict the above regarding these two points?

Line 265:

Also, please provide the model number of the nanopore membrane. Some polycarbonates are hydrophilic treated and some are not.

Line 174:

The unit of pressure should be Pa instead of bar. Check the guidelines. Please check the other areas.

Do the nanometric layers (50 nm in thickness) in S3 refer to gold? If so, it should be described as gold.

Fig. S3.2 shows that the time constant of the adsorption process is largely independent of pressure. Thus, is it not an exaggeration to state in the supporting information that "the system can be considered advection-dominated, since the Peclet Number was greater than one is an overstatement. Rather, a Peclet Number of about 10 indicates a flow dominated by diffusion, which is why a stochastic model based on the diffusion coefficient in Section V is valid.

In the expression of dissociation constants, capital K and small letter k are mixed up. Please correct overall.

Reviewer #3 (Remarks to the Author):

The manuscript by Chazot-Franguiadakis et al deals with the transport of viral particles through nano-porus membranes under pressure gradients. The authors work in a specific regime where a saturation of particle translocation frequency versus time occurs, which they attribute to jamming of viral particles inside the long nanopores. This jamming is introduced to come due to an attractive interaction between the viral capsids and the nanopore surface and the particles themselves.

Overall the paper is solid, wand warrants publication in a journal like Nature Comm. It is of sufficiently broad interest and provides an interesting contribution to both nanopore physics and physical virology. Methodology is sound and there is sufficient control experiments presented to support the authors claims. The writing is clear, but sometimes could be a bit shorter and to the point.

I would suggest several minor comments to be addressed before publication:

a) The introduction is a bit short, and could benefit with more information to position the conditions in which they work (e.g. viral concentration regimes) better in respect to prior work. Especially as certain assumptions are made to construct the models later on the paper (e.g. that the time for the viral particle to enter does not depend on the time it takes to reach the pore).

b) The authors claim that the phenomena is induced by the hydrophobic and electrostatic interactions between the viral particles (among other things). To that end they vary experimental conditions like salt concentration in the solution. I believe the data would benefit from another experiment done at a different concentration than just 150 mM. This way there is more of a variation

of the electrostatic part of the interaction. This is especially as the data on their master curve is pretty "noisy", and I worry a bit they might be "overfitting" with their model.

c) The method of how the dyes used in the work warrants more explanation, e.g. the efficiency of the chemical labeling process, and how it would impact the particles and their behaviour.

d) Please show in the supplemental a picture of how the events look like and how the clogged and non-clogged pores look like. Especially as pictures were analyzed "by eye".

e) It would be interesting to have a bit more of an outlook by the authors on where this could have impact, as well as some ideas on how/when the shape of the viruses would impact the observed effects.

f) Please explain what a "compact rim" is and why you made that assumption (before formula 1).

Answer to Reviewer omments

Reviewer #2:

The model equations constructed in this paper are in good agreement with experimental results, and the paper is considered to be of high engineering and industrial value in terms of adsorption process control in nanopore-based bionanoparticle measurements, not only for viruses but also for bacteria, extracellular vesicles, and others. The model equations are in good agreement with the phenomenon, but do not include factors related to the mechanism. In other words, the model is valid whether the adsorption mechanism of the virus on the nanopores is electrostatic, hydrophobic interaction, or other causes. This is what makes the model easy to use and valuable, but it is also what limits it to phenomenological aspects. Some of the experimental results contain findings on the adsorption mechanism of viruses on nanopores, but there seems to be a slight inconsistency in the discussion in the main body and in the Appendix, which needs to be sorted out. In summary, this paper has value in providing guidelines for analyzing and controlling the adsorption process in nanopore measurement of bionanoparticles and related measurement techniques in a broad range of applications. Please correct the comments below on points of concern.

We thank the reviewer for his positive evaluation of our work. We agree that the generality of our model make it applicable to a wide range of transported objects, but also represents a limit in order to understand the detailed mechanism of particle interactions. We have strengthened this point in the introduction of the manuscript. We have also addressed the points of concern below.

1) Although authors state that no irreversible blockage occurred in this study, the virus measurements in general electrical nanopore measurements use particle concentrations of about 10^8 - 10^{10} , and irreversible blockage commonly occurs. Any thoughts on the difference between them? Or if you can show results at such high concentrations, it would be a further contribution to nanopore research.

We thank the reviewer for his pertinent comment. Indeed, the jamming phenomenon we are investigating differs significantly from what can occur in electrical nanopore measurement. The crucial difference lies in the process reversibility. In the present case, we observed no hysteresis during the measurement of translocation frequency versus pressure/concentration, whereas in electrical detection, it typically requires pore replacement due to complete blockage [1].

Regarding concentration range, in our experiments concentration typically varied between 10^5 - 10^7 particles/mL. The maximum concentration that we have reached is 1.7×10^8 for AAV8 particles (see Figure 5 in the main manuscript). For such concentration range, we observed the same jamming behaviour. Additionally, most of the viral samples at our disposal have initial concentrations between 10^7 - 10^8 particles/mL. Overall, we have not explored concentrations exceeding 10^8 particles/mL and cannot rule out the possibility of irreversible binding within this range of concentration.

Nevertheless, in comparison electrical detection methods often operate at higher concentration (10^8 - 10^{10} particles/mL) because of their lower sensitivity. This limitation is mainly due to the use of a single pore in case of electrical detection versus 10^4 pores in parallel in our case with optical detection. More precisely, the nanoporous membrane that we used are track-etched membranes exhibiting very high pore densities (1 - 3×10^6 pores/cm²). They are made of polycarbonate originally used for water filtration and designed to exhibit a low retention of proteins contrary to SiN membrane used for electrical detection. Furthermore, we also passivate the surface with Fetal Bovin Serum (FBS) in order to further limit the potential interaction between viruses and the pores (see Supplementary Information, S3).

Our manuscript has been revised to emphasize that the reversible jamming phenomenon occurred in a concentration window that has to be determined for each virus type and depends on the interaction parameters. We have also highlighted in the text (Introduction p1-2) and in Supplementary Information (S3) that optical detection allows us to analyse low-concentration samples and emphasized the importance of membrane material and passivation in limiting virus-pore interactions.

2)Line 140: 1 - 6.108pores, where '.' is probably "x". Other similar expressions are scattered throughout. Please follow the author's instructions. Figure 2 is not cited in the main body, please cite and explain.

We thank the reviewer for his remarks. We have carefully reviewed both the main manuscript and Supplementary Information for any additional errors and made the necessary corrections, which are highlighted in red.

3) There are commercially available beads of the same size and material, differing only in surface modification. Shouldn't the verification using beads be measured with the same pore and particle size? It would be better to use only surface modification as a parameter to clearly show the effect of electrostatic interactions.

We thank the reviewer for his comment. We have performed additional experiments in order to enhance our comparison between beads with different surface properties. We relied on new polycarbonate beads coated with carboxyl groups of a larger size (radius, $R_{\text{beads}}=130$ nm) and carried out similar experiments than show in Fig. but in 400 nm pores. We performed experiments with both smaller ($R_{\text{beads}}=38$ nm) and larger ($R_{\text{beads}}=130$ nm) carboxyl beads, than the amino beads used in the article ($R_{\text{beads}}=85$ nm), using the same pore diameter (400 nm). As illustrated in Figure 1.A, a linear evolution of the translocation frequency with pressure and concentration was observed with the carboxyl nanobeads of two sizes in large pores (400 nm). The same behaviour was observed for the small carboxyl beads ($R_{\text{beads}}=38$ nm) in smaller pores (200 nm) as represented in the main manuscript (Figure 2). For amino beads (Figure 1.B), the appearance of a critical pressure, similarly to the one observed for viruses, was evidenced. However, there was no saturation at high pressures, and the increase in concentration was accompanied by an increase in translocation frequency.

This additional experiment shows that the size of the pore and of the particle does not affect the linear regime observed in the case of carboxyl beads (as long as the bead radius remains smaller than the pore radius). The difference in behaviour observed during the transport of carboxyl (Figure 1.A) versus amino (Figure 1.B) beads is therefore not due to a difference in size but rather to electrostatic and/or hydrophobic interactions.

Figure 1: A. Translocation frequency as a function of pressure for polystyrene nanobeads functionalized with carboxyl groups of two sizes at different concentrations. Linear regimes were observed (continuous color lines) contrary to viral particles. The two carboxyl beads have respective radius of: $R_{\text{beads}}=38 \pm 11$ nm, measured by DLS and $R_{\text{beads}}=130 \pm 35$ nm, provided by the supplier (*Spherotech Inc*). **B. Translocation frequency as a function of pressure for polystyrene nanobeads functionalized with amino groups at different concentrations.** Presence of a critical pressure (below which no nanobeads were observed) followed by a linear regime. Continuous color lines were fitted by $k^{-1} \frac{P_c}{P} \exp\left(\frac{P_c}{P}\right)$ (see τ_2 in equation (4) of the main manuscript). $R_{\text{beads}}=85 \pm 25$ nm, provided by the supplier (*Spherotech Inc*). **For A. and B.** Pore diameter 400 nm. Experimental errors were the standard error of the mean, and $N_{\text{replica}} > 24$ for each experimental series.

We added reference to this additional experiments in the text and the details in the Supplement to show the results from this nanobead transport measurements (see Figure S4.1 in Supplementary S4).

4) In the evaluation of the effect of electrostatic forces using beads, it would be desirable to indicate the zeta potential of the beads and nanopore membranes, since it can clearly indicate whether the electrostatic interaction between the beads and the nanopore membrane is an attractive or repulsive force. In general, the zeta potential of amino group modified beads at neutral pH is lower than that of carboxyl group modified beads, so the magnitude of the zeta potential may be more important than the positive or negative difference. In such cases, hydrophobic interactions may be dominant.

We thank the reviewer for their insightful comment.

We have carried out complementary experiments using a zetameter (Zetasizer-Nano ZS, *Malvern Panalytical*) from the Laboratoire de Chimie (ENS de Lyon) to measure the zeta potential of the different nanobead types of the same size. As depicted in Table 1, our findings revealed that the carboxyl beads ($R_{\text{beads}} = 80 \text{ nm}$) exhibited a negative zeta potential of -63.8 mV , while the amino beads ($R_{\text{beads}} = 85 \text{ nm}$) showed a zeta potential of -54.8 .

Nanobead type	Zeta potential (mV) -TE buffer
Carboxyl nanobeads ($R_{\text{beads}} = 80 \pm 20 \text{ nm}$)	-63.8 ± 9.6
Amino nanobeads ($R_{\text{beads}} = 85 \pm 25 \text{ nm}$)	-54.8 ± 2.4

Table 1: Measurement of zeta potential of the different nanobead type. Values are averaged over a minimum of 2 measurements.

The negative values for the zeta potential of both carboxyl and amino nanobeads may be explained by the low density of the amino groups grafted on the particle (10^4 amino groups per particles) which can compensate for the negative charges (carboxyl group mainly) initially present on the particle before functionalization. This measurement emphasizes that the pressure threshold observed for amino nanobeads is due to short range surface interactions.

Regarding the nanoporous membrane, measuring its zeta potential using commercial techniques is not feasible. However, scientific literature sources have provided some relevant data. Notably, R. Paoli et al. measured the zeta potential for a polycarbonate track etched membrane, corresponding to a value of -12 mV [2]. They also provided interesting information about surface charge density of such membranes (about -11 mC/m^2). The membrane used are very similar to the ones we are using therefore making a suitable comparison. Additionally, complementary studies have reported zeta potential values for track-etched membranes made from different materials, such as poly(ethylene terephthalate), which showed a value of approximately -40 mV [3].

These measurements provide valuable insight into the electrostatic interactions between the beads and the nanopore membrane. We have added this analysis in the Supplementary Information (see Table S6.1 in Supplementary S6) and modified Figure 2 in the main manuscript to highlight the global negative zeta potential of both bead types.

5) Lines 232-238: It is stated that when the virus was put in from the gold-coated side, the virus did not aggregate near the inlet and no critical pressure was generated. From this result, it is inferred that the cause of the critical pressure in the experiments in Section II is that the virus aggregated near the inlet when it was put in from the polycarbonate side. On the other hand, the description in Fig. S3.1 states that no difference is observed whether the inlet side is gold or polycarbonate. Furthermore, a closer look at Fig. S3.1 shows that the critical pressure seems to occur around 0.2-0.3 mbar. Doesn't this contradict the above regarding these two points?

We agree with the remark of the reviewer and we understand that our explanation may have been confusing.

In our study, "exit experiments" referred to experiments conducted by observing the exit of virus translocation, where gold was positioned on the exit side and polycarbonate at the entry side. Conversely, "entry experiments" referred to those performed by observing the entry of viruses into the nanopores, with gold located on the entry side and polycarbonate at the exit side. Indeed, gold is always needed on the observation side to take advantage of the Zero Mode Waveguide effect. This distinction is illustrated in Figure 2 below.

Figure 2: Diagram explaining the different configurations of the experiments. (Left) Exit experiment (with one side gold); (Middle) Exit experiment (with two sides gold); (Right) Entry experiment (with one side gold). The configuration used conventionally is the one on the left.

Our intention was to state that the difference between entry versus exit experiments (i.e. the absence of a critical pressure for entry experiments) implies that the clog is located inside the nanopore channel. Indeed, during both entry and exit experiments we did not visually observe any clogged particles. Moreover, additional experiments in Supplementary Information highlighted that no difference was observed when we performed exit experiments with either gold or polycarbonate at the entry side (see Fig S4.2 in Supplementary S4).

To provide further clarity on this matter, we have revised the Supplementary Information (S4) and included a diagram illustrating this concept (Fig S4.2.A).

6) Line 265: Also, please provide the model number of the nanopore membrane. Some polycarbonates are hydrophilic treated and some are not.

We thank the reviewer for bringing to our attention the model number of the nanoporous membrane is missing. In this study, we relied on track-etched membrane: "Whatman Nucleopore (GE Healthcare) in polycarbonate" of four different pore diameters: 80 nm (reference WHAT10419306); 100 nm (reference WHAT10419506); 200 nm (reference WHAT10417006) and 400 nm (reference WHAT10417106).

We have added this information in the Supplementary Information (S1).

7) Line 174: The unit of pressure should be Pa instead of bar. Check the guidelines. Please check the other areas.

We thank the reviewer for his comment. We have converted the pressure units from bar to Pa in both the main manuscript and the Supplementary Information. The modifications have been highlighted in red for easy tracking of the changes.

8) Do the nanometric layers (50 nm in thickness) in S3 refer to gold? If so, it should be described as gold.

We thank the reviewer for his comment. Indeed, the nanometric layer (of 50 nm in thickness) described in initially S3 (now S4) corresponds to gold. We have clarified this in the text.

9) Fig. S3.2 shows that the time constant of the adsorption process is largely independent of pressure. Thus, is it not an exaggeration to state in the supporting information that "the system can be considered advection-dominated, since the Peclet Number was greater than one is an overstatement. Rather, a Peclet Number of about 10 indicates a flow dominated by diffusion, which is why a stochastic model based on the diffusion coefficient in Section V is valid.

We thank the reviewer for his comment. If we take into considerations all the pressures, pore size and both viral particle and bead size, we found a Peclet between: $2 < Pe < 400$. More precisely, for viral particles in the saturation regime: $10 < Pe < 400$.

Therefore, in the simple case of a competition between advection and diffusion, advection would dominate. This is the case for negatively charged nanobeads (see Fig. 2.A in the article). However, when additional interactions such as particle aggregation, binding and unbinding to the pore surface are present, this consideration should be completed by an estimate of the ratio of binding rate over advection rate. For example, the competition between binding and advection can be estimated with the advection-binding number ($\frac{k_{on}^{pore} L}{v}$). In the geometry of our experiments, this number is ranging between 0.05 and 3 for the pressure range used in our experiments showing that we operate at a cross over between binding and advection. Unfortunately, the determination of all the dynamical parameters (binding, unbinding to the pore and the other viruses) is inaccessible with our experiments.

Overall, we have modified the manuscript to highlight this complex competition between advection, diffusion and binding in Supplementary Information (S5) and added information in S4 (Figure S4.4, initially S3.2).

10) In the expression of dissociation constants, capital K and small letter k are mixed up. Please correct overall.

We thank the reviewer for bringing to our attention this confusion. We have carefully reviewed both the main manuscript and Supplementary Information for any additional errors and made the necessary corrections, which are highlighted in red.

Reviewer #3:

The manuscript by Chazot-Franguiadakis et al deals with the transport of viral particles through nanoporous membranes under pressure gradients. The authors work in a specific regime where a saturation of particle translocation frequency versus time occurs, which they attribute to jamming of viral particles inside the long nanopores. This jamming is introduced to come due to an attractive interaction between the viral capsids and the nanopore surface and the particles themselves.

Overall the paper is solid, and warrants publication in a journal like Nature Comm. It is of sufficiently broad interest and provides an interesting contribution to both nanopore physics and physical virology. Methodology is sound and there is sufficient control experiments presented to support the authors claims. The writing is clear, but sometimes could be a bit shorter and to the point.

We thank the reviewer for his positive evaluation of our work. We have carefully reviewed both the main manuscript and Supplementary Information aiming to make it more concise when necessary. We have also addressed the points of concern below.

I would suggest several minor comments to be addressed before publication:

1) The introduction is a bit short, and could benefit with more information to position the conditions in which they work (e.g. viral concentration regimes) better in respect to prior work. Especially as certain assumptions are made to construct the models later on the paper (e.g. that the time for the viral particle to enter does not depend on the time it takes to reach the pore).

We thank the referee for bringing attention to this point.

We have completed the state of the art by providing comprehensive insights into the concentration regime used in this manuscript compared to the concentration range typical of other methods, in particular nanopore electrical detection methods. Indeed, the jamming phenomenon we are investigating differs significantly from what can occur in electrical nanopore measurement. The crucial difference lies in the process reversibility. In the present case, we observed no hysteresis during the measurement of translocation frequency versus pressure/concentration, whereas in electrical detection, it typically requires pore replacement due to complete blockage [1].

This limitation is partly due to the fact that electrical methods require high viral concentrations, typically around 10^8 - 10^{10} particles/mL because of their low sensitivity. This is mainly due to the use of a single pore (no parallelisation of the system) and the membrane material (Silicium Nitride, SiN) that do not limit protein interactions.

In our experiments, concentration typically varied between 10^5 - 10^7 particles/mL. Indeed, we can achieve lower detection limit thanks to the nanoporous membrane used, that are tracked etched membranes exhibiting very high pore densities (1 - 3×10^6 pores/cm²), representing 10^4 pores on the observed surface. The nanoporous membranes are also made of polycarbonate originally used for water filtration and designed to exhibit a low retention of proteins. Furthermore, we also passivated the surface with Fetal Bovin Serum (FBS) in order to further limit the potential interaction between viruses and the pores (see Supplementary Information, S1).

Furthermore, we have also highlighted the fact that the conditions used in this article were coherent with the concentration range encountered in biological contexts. Indeed, the viral concentrations encountered in biological conditions, such as patient biofluids, can vary from 10^3 to 10^7 particles/mL, depending on various factors such as the type and stage of infection [4]. This alignment underlines the relevance and applicability of our findings to understanding biological mechanisms and strengthens the translational potential of our research.

We have highlighted those point in the state of the art of the main manuscript and also in Supplementary Information (S3).

Additionally, we have also completed the manuscript to help in the understanding of further assumptions that were made to construct our jamming model. Especially, we have added references regarding the fact that the time for the viral particle to enter the pore does not depend on the time it takes to reach the pore. Indeed, Auger et al. have shown that in the case of double stranded DNA transport through nanopores, there is the presence of a confinement barrier limiting the transport of DNA (suction model) [5,6]. We have also used this model in the manuscript when characterizing the state of the clog (see Section III in the main manuscript). Here, the barrier is limited not by a confinement energy barrier but by the accessibility of the site as in Langmuir isotherm models. Therefore, the translocation time is dominated by this barrier while the time to reach the pore is negligible.

We have added this information in Section V of the main manuscript (p7).

2) The authors claim that the phenomena is induced by the hydrophobic and electrostatic interactions between the viral particles (among other things). To that end they vary experimental conditions like salt concentration in the

solution. I believe the data would benefit from another experiment done at a different concentration than just 150 mM. This way there is more of a variation of the electrostatic part of the interaction. This is especially as the data on their master curve is pretty "noisy", and I worry a bit they might be "overfitting" with their model.

We thank the reviewer for their insightful remarks. We have carried out complementary experiments on HIV (VLP) particles with concentrations of NaCl of 150, 300 and 600 mM. As depicted in Figure 3, our findings revealed that the addition of NaCl to HIV solutions led to an increase in the translocation frequency higher and a less pronounced saturation. The behaviour appeared independent of NaCl concentration, as the obtained curves were similar for 150, 300 and 600 mM. In this case, the transport of viruses was facilitated by electric charge screening that prevented electrostatic interactions of viruses with each other or with the pore.

Figure 3: HIV translocation with different NaCl concentrations. Evolution of normalized translocation frequency as a function of normalized pressure for HIV particles: naked or with NaCl (150 mM, 300 mM and 600 mM). The translocation frequency was higher with salt, compared to naked membrane. The saturation was also less marked in this case. Nevertheless, it did not seem to depend on NaCl concentration.

We have added a reference to these additional experiments in the main manuscript and provided detailed information in the Supplementary Information (see Figure S4.6 in Supplementary S4).

3) The method of how the dyes used in the work warrants more explanation, e.g. the efficiency of the chemical labeling process, and how it would impact the particles and their behaviour.

We thank the reviewer for his comment. We have provided complementary information about the labelling methodologies.

To visualize objects emerging from nanopores using Zero-Mode Waveguide technology, they must first be fluorescently labelled. This labelling process varies depending on the target of the fluorescent label. Two labelling strategies have been employed: targeting of genetic material (DNA, RNA) using YOYO-1 or fluorescent labelling inherent to virus production (e.g Gag-GFP), as illustrated in Figure 1.A of the main manuscript. The two markers were chosen to have excitation and emission wavelengths compatible with the experimental set-up (see Supporting Information S1 & S2). Moreover, the fluorescent markers used do not modify the properties of the viral particles. They are passive markers, with no interaction with the nanopores, and do not alter the translocation of the viral particles.

More precisely, YOYO-1 is used to fluorescently labelled genetic material (DNA or RNA). It is a green intercalant belonging to the cyanine family. Alone in solution, YOYO-1 has a low quantum yield and can be considered as almost non-fluorescent, whereas when bound to DNA/RNA with which it forms a complex, its yield increases 1000-fold [7-9]. Its fluorescence spectrum, indicates excitation and emission wavelengths of 491 nm and 509 nm respectively (Figure 4.B). YOYO-1, whose chemical structure is shown in Figure 4.A, binds to DNA by bis-intercalation of its two chromophore units.

Figure 4: Fluorophore YOYO-1. A. Chemical formula of YOYO-1. B. Emission and excitation spectra of YOYO-1, produced with SpectraViewer.

YOYO-1 is purchased in 200 μL solution at 1 mM in DMSO from Molecular probes. We optimized the YOYO-1 to DNA ratio to ensure DNA chain saturation (1 intercalant per 4 DNA bases) without increasing background fluorescence. Moreover, the addition of YOYO-1 leads to an approximately 38% increase in DNA contour length when the chain is saturated with YOYO-1. This corresponds to a new length per base of around 0.47 nm when YOYO-1 is saturated, compared with 0.34 nm without YOYO-1 [6-8]. Nevertheless, the use of YOYO-1 does not alter the persistence length of DNA. Furthermore, the value of the equilibrium dissociation constant between DNA and YOYO-1 ranges from 700 pM to 12 nM, equivalent to a free enthalpy of reaction of around 18-21 $k_B T$ [10,12].

We have evidenced that YOYO-1 is able to label DNA inside AAV and HBV viruses, by effectively penetrating their viral capsids. One possible mechanism for this penetration is diffusion through pores naturally present in the capsid. Importantly, the labelling occurring inside the capsid does not affect the interaction of the virus with the nanopore. Furthermore, we also conducted a comparative analysis of images from free DNA in solution versus DNA encapsulated within viral capsids (Figure 5). The spatial distribution of fluorescence signals allows us to distinguish between these two conditions.

Figure 5: Images corresponding to translocation exit event of dsDNA and HBV labelled using YOYO-1. A. dsDNA correspond to λ -DNA (linear double stranded DNA of 48 502 pb). B. HBV capsids contain double stranded DNA partially circular of 3200 pb. Pores 200 nm. Image size: 15 μm . Frame rate: 33 fps.

Regarding inherent fluorescent labelling, we relied on Gag-GFP for both HIV (VLP) and MLV particles. More precisely, for HIV (VLP) and MLV, fluorescent labelling is achieved using GFP fused to the C-terminal position of the Gag structural protein. In the case of the full-length MLV virus, GFP is incorporated into the center of the Gag polyprotein. The fluorescence spectrum of GFP is shown in Figure 6, with excitation and emission wavelengths of 488 nm and 507 nm respectively. The natural GFP protein has 238 amino acids and a molecular mass of 27 kDa. Despite this high molar mass, the addition of GFP to Gag protein does not alter its assembly and release properties [13,14].

Figure 6: GFP, Green Fluorescent Protein. A. Structure of the natural GFP protein from the jellyfish *Aequorea victoria*, taken from [15]. B. Emission and excitation spectra of GFP, produced using with SpectraViewer. C. Images corresponding to translocation exit event of HIV (VLP) labelled using Gag-GFP. Pores 400 nm. Image size: 15 μ m. Frame rate: 33 fps.

We have added this additional description of the fluorescent labelling in the Supplementary Information (S1, Figures S1.1 & S1.2) as well as references in the main manuscript.

4) Please show in the supplemental a picture of how the events look like and how the clogged and non-clogged pores look like. Especially as pictures were analyzed "by eye".

We thank the referee for his comment. Indeed, while our focus lies on observing the exit/entry of viral particles from/in the pores, it is crucial to acknowledge the limitation regarding our ability to investigate the interior of a clogged pore. At present, we lack a method capable of effectively probing the internal structure of a jammed pore. We acknowledge that our previous explanation may have caused confusion, and we tried to clarify that in our manuscript. What we intended to convey was that despite examining both entry and exit events, we did not detect the presence of aggregated particles or clogs at pore exit or entrance. This absence led us to assume that any potential blockages likely form within the pores themselves. To complement our findings, we include snapshots of HIV (VLP) particles exiting and entering nanopores (Figure 7). These snapshots provide valuable visual insights into the behaviour of viral particles within our exit versus entry experimental setup.

Figure 7: Images corresponding to the translocation of HIV (VLP) labelled using Gag-GFP. A. Exit experiment. B. Entry experiment. Each line (A or B) correspond to a single event. Pores 400 nm grafted with poly(2-methyl-2-oxazoline)s. Image size: 15 μ m. Frame rate: 33 fps.

We have added this additional description of the fluorescent labelling in the Supplementary Information (S4, Figure S4.3) as well as references in the main manuscript.

5) It would be interesting to have a bit more of an outlook by the authors on where this could have impact, as well as some ideas on how/when the shape of the viruses would impact the observed effects.

We thank the reviewer for this insightful remark. We have tried to develop the potential impacts and application perspectives of the soft jamming phenomenon described in this manuscript.

First, it offers a new approach for characterizing surface states, providing a valuable insight for studying the influence on drugs on viral particle and their interactions. Similarly, it can also be used to study other type of biological particle which have similar size and biological composition of viruses but that exhibit different surface receptors as extracellular vesicles. Such objects offer promising approaches for both diagnostic and therapeutic (as vectors) purposes.

Overall, the jamming phenomenon, evidenced in this article, enables to engineer controlled aggregation of patchy nanoparticles under flow confinement, promising advancements in materials science and biotechnology.

We have added these perspectives in Section VI- Conclusion, of the main manuscript.

6) Please explain what a "compact rim" is and why you made that assumption (before formula 1).

We thank the referee for his comment. As "compact rim" may be confusing, we have replaced it by "compact layer" both in the manuscript and Supplementary Information.

The reference to a compact layer meant that we assumed that viruses formed a layer all over the pore, leaving a cylindrical space at the center of the pore which exhibited a radius smaller than the empty pore radius, as schematized in Figure 8.

Moreover, the compact layer hypothesis is necessary to extract the virus layer thickness from the critical pressure (P_c) originating from the suction model (Section III of the main manuscript). This is a simplifying hypothesis to get information on the crowding of the pore. However, we totally agree with the referee that more complex architecture of the virus aggregate may be involved.

Figure 8: Diagram of the simplified architecture of the virus clog inside the pore. R is the radius of the empty pore while R_e corresponds to the virus layer thickness.

We have added this in the main manuscript (Figure 3.C).

References:

- [1] *Selective detections of single-viruses using solid-state nanopores*, A. Arima et al., *Scientific Reports*, Vol 8 (2018).
- [2] *Layer-by-layer modification effects on a nanopore's inner surface of polycarbonate track-etched membranes*, R. Paoli et al., *RSC Advances*, Vol 10, 35930 (2020).
- [3] *Streaming Potential in Cylindrical Pores of Poly(ethylene terephthalate) Track-Etched Membranes: Variation of Apparent ζ Potential with Pore Radius*, P. Dejardin et al., *Langmuir*, Vol 21, 10, 4680-4685 (2005).
- [4] *Virological assessment of hospitalized patients with COVID-2019*, R. Wölfel, et al., *Nature* 581, 465–469 (2020).
- [5] *Injection Threshold for a Statistically Branched Polymer inside a Nanopore*, C. Gay, et al., *Macromolecules* 29, 8379 (1996).
- [6] *Zero-Mode Waveguide Detection of Flow-Driven DNA Translocation through Nanopores*, T. Auger et al., *Physical Review Letter*, 113, Iss 2 (2014).
- [7] *Bis-intercalating asymmetric cyanine dyes : properties and applications*, H. S. Rye, et al., *Nucleic acids research*, vol. 20, no. 11, pp. 2803–2812 (1992).
- [8] *Base-content dependence of emission enhancements, quantum yields, and lifetimes for cyanine dyes bound to double-strand DNA : Photophysical properties of monomeric and bichromophoric DNA stains*, T. L. Netzel, et al., *Journal of Physical Chemistry*, vol. 99, no. 51, pp. 17936–17947 (1995).
- [9] *Ultrafast excited state dynamics of DNA fluorescent intercalators: New insight into the fluorescence enhancement mechanism*, A. Fürstenberg, et al., *Journal of the American Chemical Society*, vol. 128, no. 23, pp. 7661–7669 (2006).
- [10] *The kinetics of YOYO-1 intercalation into single molecules of double-stranded DNA*, M. Reuter and D. T. Dryden, *Biochemical and Biophysical Research Communications*, vol. 403, no. 2, pp. 225–229 (2010).
- [11] *Mechanical and structural properties of YOYO-1 complexed DNA*, K. Günther, M. Mertig, and R. Seidel, *Nucleic Acids Research*, vol. 38, no. 19, pp. 6526–6532 (2010).
- [12] *Effect of YOYO-1 on the mechanical properties of DNA*, B. Kundukad, J. Yan, and P. S. Doyle, *Soft Matter*, vol. 10, no. 48, pp. 9721–9728 (2014).
- [13] *NXF1 and CRM1 nuclear export pathways orchestrate nuclear export, translation and packaging of murine leukaemia retrovirus unspliced RNA*, M. Mougél, et al., *RNA Biology*, vol. 17, pp. 528–538, (2020).
- [14] *Protein transduction from retroviral Gag precursors*, C. Voelkel, et al., *Proceedings of the National Academy of Sciences*, vol. 107, pp. 7805–7810 (2010).
- [15] *Crystal structure of the Aequorea Victoria green fluorescent protein*, M. Ormo, et al., *Science*, vol. 273, pp. 1392–1395 (1996).

REVIEWERS' COMMENTS

Reviewer #2 (Remarks to the Author):

The manuscript has been revised well. I think this manuscript will be acceptable after some corrections have been done.

General Considerations

The Soft Jamming model consists of two items: entry duration and translocation duration. In the current structure of the paper, the two effects are intermingled in the experimental results, which at first glance may read as contradictory in some parts. Once one reads through to section V, the relationship and consistency between the two effects is created. Since it is not enough to modify the overall structure, it would be easier to read the text if the two dominant effects were forewarned at the beginning of the main text.

Minor corrections

1) Line 69 – 77:

“This is mainly due to the use of a single pore (no parallelisation of the system) and the membrane material (Silicium Nitride, SiN) that do not limit protein interactions.”

- You should cite references to support your argument.

- 'Parallelisation' is British English, so please be careful to use either British or American English throughout the paper.

- Wouldn't it be better to use Silicium -> Silicon?

2) Line 93 - 95

“Here, a different pressure and concentration regime (comparable to biological fluid concentrations) was explored”

The authors should cite references that show the pressure and virus particle concentration in the BIOLOGICAL fluid claimed here.

3) Fig. 3 and others

- "...for HIV particles (95.105 particles/mL)..."

Line 506

- $4/3\pi R^3 \times \pi R^2 L$

In some cases, arithmetic operators are written as 'x' and '!'.
Please unify them with 'x'.

4) Line 198

There is no explanation for "plateau frequency", please explain it.

5) Which of the following two P_c is correct?

Line 193 For HIV particles and 200 nm diameter pores, " $P_c = 50 \pm 10 \text{ Pa}$ ".

Line 259 (for HIV particles and 200 nm diameter pores, " $P_c = 10^{-3} \text{ Pa}$ ")

Line 272

"a fixed pressure of 800 Pa (frequency saturation regime)", is it correct to assume that this is the case where the Cis side is the inlet? Since the critical frequency $P_c=103\text{Pa}$ (line 260), isn't 800Pa before "frequency saturation regime"?

In Fig. S4.4, is it correct to assume that the 'absolute number' of particles passing through increases with higher pressure, although it is not readable from the figure because the vertical axis is normalized? Perhaps f_0 and f_∞ are expected to be larger at higher pressures.

Line 337

" $Re = 0.5R$ for all viruses and pore diameters." but in Fig. 3C, $Re = 0.48R$. Please unify the effective digits.

Is it correct to understand that this means that smaller viruses are adsorbed in more layers compared to larger viruses?

In addition, Fig. S4.4 claims that the time constant of the clogging process does not change at 800Pa or 1600Pa, on the other hand, the clogging is released at 10^4 Pa. These suggest that the interaction between the virions cannot be neglected and that the shear force from the flow is influential, is it possible to make a discussion that results in the relationship of $Re = 0.5R$?

Regarding the newly added Supporting information S4 Additional Experiments.

In the experiment in Supporting information S4, Fig. S4-2, the presence of a critical pressure is observed regardless of whether the inlet or outlet is gold or polycarbonate (Fig. S4-2B). On the other hand, Fig. 3A shows no critical pressure when the inlet side is gold. Is this not a contradiction?

It would be better to include the orientation of the time lapse in Fig. S4.3.

In the second line of the section Transient State of Clog Formation in Supporting information S4

Fig. 2B -> Fig. 3B?

The K in K_{clogd} is mixed case. It should be unified to one of them, probably capitalized.

Line 506 & Fig. S4.3

Based on the equation here, for example, with a nanopore diameter of 400 nm and HIV diameter of 150 nm, about 400 HIV particles are adsorbed on the inner wall of the nanopore, and if we burst at 10^4 Pa, will we observe the particles being ejected at once?

This may help to strengthen the validity of the model in this paper.

Line 519

Define what k_{diff} represents.

Reviewer #3 (Remarks to the Author):

I am satisfied with the response of the authors. I would recommend publishing the paper.

Reviewer #2 (Remarks to the Author):

The manuscript has been revised well. I think this manuscript will be acceptable after some corrections have been done.

We thank the reviewer for his positive evaluation of our work.

General Considerations

The Soft Jamming model consists of two items: entry duration and translocation duration. In the current structure of the paper, the two effects are intermingled in the experimental results, which at first glance may read as contradictory in some parts. Once one reads through to section V, the relationship and consistency between the two effects is created. Since it is not enough to modify the overall structure, it would be easier to read the text if the two dominant effects were forewarned at the beginning of the main text.

Minor corrections

1) Line 69 – 77: "This is mainly due to the use of a single pore (no parallelisation of the system) and the membrane material (Silicium Nitride, SiN) that do not limit protein interactions."

- You should cite references to support your argument.

- 'Parallelisation' is British English, so please be careful to use either British or American English throughout the paper.

- Wouldn't it be better to use Silicium -> Silicon?

We thank the reviewer for his remarks. We have carefully reviewed and corrected those elements, which are highlighted in red.

2) Line 93 – 95: "Here, a different pressure and concentration regime (comparable to biological fluid concentrations) was explored". The authors should cite references that show the pressure and virus particle concentration in the BIOLOGICAL fluid claimed here.

We thank the reviewer for his remarks. We have added the corresponding reference in red.

3) Fig. 3 and others: ".....for HIV particles (95.10⁵ particles/mL)...."

Line 506: - $4/3\pi R^3 \cdot \pi R^2 L$

In some cases, arithmetic operators are written as 'x' and '·'.

Please unify them with 'x'.

We thank the reviewer for his remarks. We have carefully reviewed and unified operators as "x" and those changes are highlighted in red.

4) Line 198: There is no explanation for "plateau frequency", please explain it.

We thank the reviewer for his remarks. We have added an explanation in the text.

5) Which of the following two P_c is correct?:

Line 193 For HIV particles and 200 nm diameter pores, " $P_c = 50 \pm 10$ Pa".

Line 259 (for HIV particles and 200 nm diameter pores, " $P_c = 10^{-3}$ Pa").

We thank the reviewer for his remark and we understand that this point may have been confusing. The first $P_c = 50 \pm 10$ Pa, line 193, corresponds to HIV particles in 200 nm diameter pores and **exit experiments**. In our study, "exit experiments" refer to experiments conducted by observing the exit of virus translocation, where gold is positioned on the exit side and polycarbonate at the entry side.

The second $P_c = 10^{-3}$ Pa, line 259, corresponds also to HIV particles in 200 nm diameter pores but for **entry experiments**. "Entry experiments" refer to experiments performed by observing the entry of viruses into the nanopores, with gold located on the entry side and polycarbonate at the exit side. This distinction is illustrated in Fig. S4.2 and we have added clarification in the text as highlighted in red.

Line 272: "a fixed pressure of 800 Pa (frequency saturation regime)", is it correct to assume that this is the case where the Cis side is the inlet? Since the critical frequency $P_c=10^3$ Pa (line 260), isn't 800Pa before "frequency saturation regime"?

We thank the referee for the comment. We believe there has been some confusion: the critical pressure, mentioned in line 259, is $P_c=10^{-3}$ Pa, not 10^3 Pa. Additionally, the experiments referenced in line 272 are exit experiments. In this case, the critical pressure is around 50 Pa, and 800 Pa falls within the frequency saturation regime, as shown in Fig. S5.1.

In Fig. S4.4, is it correct to assume that the 'absolute number' of particles passing through increases with higher pressure, although it is not readable from the figure because the vertical axis is normalized? Perhaps f_0 and f_∞ are expected to be larger at higher pressures.

We thank the referee for his insightful remark. The 'absolute number' of particles passing through the pore is indeed increasing with higher pressures. More precisely, the number of events at $t=0$ and $t=\infty$ are for 800 Pa: $N_\infty=8$, $N_0=20$ and for 1600 Pa: $N_\infty=14$, $N_0=26$. This aspect was not explored further in this article, but could be the subject of future investigation with complementary experiments.

Line 337: " $Re = 0.5R$ for all viruses and pore diameters." but in Fig. 3C, $Re = 0.48R$. Please unify the effective digits.

We thank the reviewer for his remarks. We have corrected those elements.

Is it correct to understand that this means that smaller viruses are adsorbed in more layers compared to larger viruses?

We agree with the referee that based on these results, we can infer that a possible mechanism is that smaller viruses are absorbed into more layers than larger viruses.

In addition, Fig. S4.4 claims that the time constant of the clogging process does not change at 800Pa or 1600Pa, on the other hand, the clogging is released at 10^4 Pa. These suggest that the interaction between the virions cannot be neglected and that the shear force from the flow is influential, is it possible to make a discussion that results in the relationship of $Re = 0.5R$?

In our opinion these results do not imply that the interactions between the virion are dominant and we don't think that we can make, at this stage, a link between the structure of the adsorbed layer and the shear flow necessary to detach the virus.

Regarding the newly added Supporting information S4 Additional Experiments. In the experiment in Supporting information S4, Fig. S4-2, the presence of a critical pressure is observed regardless of whether the inlet or outlet is gold or polycarbonate (Fig. S4-2B). On the other hand, Fig. 3A shows no critical pressure when the inlet side is gold. Is this not a contradiction?

We thank the reviewer for his comment. On one hand, the experiments exhibited in Fig. S4-2, correspond both to '**exit experiments**' which refer to experiments conducted by observing the exit of virus translocation. In the 'one side gold' case, the gold layer was only positioned at exit while in the 'two sides gold', gold layers were positioned at entry and exit.

On the other hand, Fig. 3A corresponds to 'entry experiments,' which involve observing the entry of viruses into the nanopores. In these experiments, gold is located on the entry side and polycarbonate on the exit side, with the system operating in suction mode.

Here, the main point is that the critical pressure is independent of the presence of the gold layer; instead, it depends on whether we performed entry or exit experiments (observing viruses entering or exiting the nanopores). This difference can account for the presence of a clog inside the nanopore.

It would be better to include the orientation of the time lapse in Fig. S4.3.

We thank the reviewer for his comment. We have added a time lapse in Fig. S4.3.

Figure S4.3: Images corresponding to the translocation of HIV (VLP) labelled using Gag-GFP. (up) Exit experiment. (down) Entry experiment. Each line (up or down) correspond to a single event. Pores 400 nm grafted with poly(2-methyl-2-oxazoline)s. Image size: 15 μm . Frame rate: 33 fps.

In the second line of the section Transient State of Clog Formation in Supporting information S4 Fig. 2B -> Fig. 3B?

We thank the reviewer for his comment and corrected this mistake.

The K in K^{clog_d} is mixed case. It should be unified to one of them, probably capitalized. We thank the reviewer for his comment. We have corrected those elements.

Line 506 & Fig. S4.3: Based on the equation here, for example, with a nanopore diameter of 400 nm and HIV diameter of 150 nm, about 400 HIV particles are adsorbed on the inner wall of the nanopore, and if we burst at 10^4 Pa, will we observe the particles being ejected at once? This may help to strengthen the validity of the model in this paper.

When we apply 10^4 Pa the virus speed is increased drastically and it is impossible with our detection system to quantify the amount of virus that are pushed outside of the pore by this increase of pressure.

Line 519: Define what k_{diff} represents.

We thank the reviewer for his comment and simplified this point.

Reviewer #3 (Remarks to the Author):

I am satisfied with the response of the authors. I would recommend publishing the paper.

We thank the reviewer for his positive evaluation of our work.